# Quality of Life and Patient-Reported Outcomes Following Proton Therapy for Oropharyngeal Carcinoma: A Systematic Review

**DOI:** 10.3390/cancers15082252

**Published:** 2023-04-12

**Authors:** Noorazrul Yahya, Hanani Abdul Manan

**Affiliations:** 1Diagnostic Imaging and Radiotherapy, Center for Diagnostic, Therapeutic and Investigative Studies (CODTIS), Faculty of Health Sciences, National University of Malaysia, Jalan Raja Muda Aziz, Kuala Lumpur 50300, Malaysia; 2Functional Image Processing Laboratory, Department of Radiology, Universiti Kebangsaan Malaysia Medical Centre, Cheras, Kuala Lumpur 56000, Malaysia; hanani@ukm.edu.my

**Keywords:** proton therapy, oropharyngeal carcinoma, patient-reported outcomes, quality of life

## Abstract

**Simple Summary:**

Proton therapy is a potentially attractive option to treat oropharyngeal cancers due to better dose distribution. We aimed to synthesize the quality of life (QOL) and patient-reported outcomes (PROs) following treatment. We found that studies frequently demonstrated the advantages of proton therapy compared to photon therapy in terms of QOL and PROs; however, biases due to the non-randomized nature of the studies may require confirmation in phase III randomized controlled trials.

**Abstract:**

Background: Complex anatomy surrounding the oropharynx makes proton therapy (PT), especially intensity-modulated PT (IMPT), a potentially attractive option due to its ability to reduce the volume of irradiated healthy tissues. Dosimetric improvement may not translate to clinically relevant benefits. As outcome data are emerging, we aimed to evaluate the evidence of the quality of life (QOL) and patient-reported outcomes (PROs) following PT for oropharyngeal carcinoma (OC). Materials and Methods: We searched PubMed and Scopus electronic databases (date: 15 February 2023) to identify original studies on QOL and PROs following PT for OC. We employed a fluid strategy in the search strategy by tracking citations of the initially selected studies. Reports were extracted for information on demographics, main results, and clinical and dose factor correlates. Quality assessment was performed using the NIH’s Quality Assessment Tool for Observational Cohort and Cross-Sectional Studies. The PRISMA guidelines were followed in the preparation of this report. Results: Seven reports were selected, including one from a recently published paper captured from citation tracking. Five compared PT and photon-based therapy, although none were randomized controlled trials. Most endpoints with significant differences favored PT, including xerostomia, cough, need for nutritional supplements, dysgeusia, food taste, appetite, and general symptoms. However, some endpoints favored photon-based therapy (sexual symptoms) or showed no significant difference (e.g., fatigue, pain, sleep, mouth sores). The PROs and QOL improve following PT but do not appear to return to baseline. Conclusion: Evidence suggests that PT causes less QOL and PRO deterioration than photon-based therapy. Biases due to the non-randomized study design remain obstacles to a firm conclusion. Whether or not PT is cost-effective should be the subject of further investigation.

## 1. Introduction

Among patients treated for oropharyngeal carcinoma (OC), acute and late treatment-related sequelae, including xerostomia, dysphagia, and dysgeusia, remain challenging [1,2,3]. These consequences of treatment may impact the quality of life (QOL), including difficulties in communication, nutritional intake enjoyment, and social contact [3]. Maintaining good QOL following OC is especially important due to improved survivorship related to better treatment regimens and the increased prevalence of human papillomavirus (HPV)-related squamous cell carcinoma, which disproportionately impacts young and physically fit individuals [4]. Thus, improving therapies to mitigate these life-altering side effects is paramount.

Numerous studies have empirically demonstrated the impact of photon dose on salivary glands [5,6,7], swallowing muscles [8,9], the oral cavity [3], and other normal tissues to patient-reported outcomes (PROs) and QOL measures in photon-based therapy. Proton therapy (PT) offers several advantages over photon therapy, including the elimination of exit dose and the possibility to tailor the dose resulting from the physical characteristics of the beams with Bragg peaks [10]. Several dose comparisons and clinical studies have shown significant normal tissue dose reductions favoring PT [11,12,13].

While dosimetric analyses have proven the benefits of PT based on dosimetric advantage [14,15], PT is not without its challenges [16]. Deviations of the given dose distribution from the intended distribution due to uncertainties arising from intrafraction motion and patient set-up error are significantly more prominent with pencil beam scanning techniques. However, several clinical approaches have been suggested to minimize the uncertainties. Volumetric repainting can statistically average the motion effects by allowing the energy layers of the proton beam to be delivered more than once through the whole target volume [17,18]. In addition, 4D robust optimization of the dose, incorporating data on time structures of delivery and intrafractional motion, has been shown to create robust treatment plans against interplay effects [19,20]. It should be noted that set-up error is not unique to proton therapy which can be lessened with appropriate volumetric image guidance.

Furthermore, the range of uncertainties as a systematic error can be between 3 and 3.5%. Combining these two sources of errors results in the radiological path length of proton beams, which differs from the intended length. While the simulation of radiation is highly accurate, the modeling of the human body based on CT images of the patient is not similarly precise due to the calibration uncertainties between the Hounsfield unit (HU) values and the proton stopping powers and intra- and interfraction variations in anatomy [21]. Beddok et al., in a review, have outlined the challenges and how they can be at least partially overcome [21]. The systematic error from range uncertainties and HU stopping power uncertainties can be accounted for in the optimization of treatment plans, or proper margins at the beam direction can be applied, which may also aid in smearing other unexpected uncertainties [22,23]. These errors are relevant to normal tissue effects as improved dose distribution may not necessarily improve PRO and QOL measures, especially in the presence of predictive clinical parameters, including baseline symptoms and other comorbidities [24,25]. The relationship between dose distribution and PROs and QOL may also depend on other technical factors, including how the treatment is monitored through cone beam computed tomography (CBCT) and whether adaptive radiotherapy is applied when significant dose deviations are observed [26].

There appear to be a few reports presenting the actual impact of PT on PROs and QOL. Verma et al., in 2018, performed a systematic review of PROs and QOL following proton therapy and found only one study for head and neck cancers [27]. The limited availability of PT in the past may be the reason for the sparse empirical evidence. Furthermore, some institutions prefer to assess toxicities based on physician-reported measures, probably due to the historical emphasis on these measures. However, as more centers offer PT and there is increasing interest in measuring outcomes that are more patient-centered and connect to patients more meaningfully, we may see more clinical results that involve PROs and QOL. In this study, we aimed to evaluate the evidence around QOL and PRO measures following PT for OC, with a secondary aim to compare the outcomes of PT and photon-based therapy.

## 2. Materials and Methods

### 2.1. Systematic Review Protocol and Eligibility Criteria

The systematic review reporting guideline established by Preferred Reporting Items for Systematic Reviews and Meta-Analyses (PRISMA) was utilized [28] (Appendix A). Original research manuscripts were evaluated for inclusion or exclusion based on PICOTS criteria (Appendix A). The PICOTS framework enables systematic inclusion of articles based on patient population (patients treated for oropharyngeal carcinoma), intervention (proton therapy), comparison (none required, but the comparison was made to baseline or alternative treatment, i.e., photon-based therapy), outcome (patient-reported outcomes or quality of life measures), time (any time frame divided into acute, sub-acute, late (within one year) and late (after one year), and study design (original research with at least 30 patients to enable statistical comparisons and to filter case series and case reports). Reports fulfilling all six criteria were included. Excluded studies were reported based on the first PICOTS criterion not being met. The protocol of this systematic review was not published elsewhere and was not registered.

### 2.2. Search Strategy and Selection Process

Electronic databases (National Center for Biotechnology Information (PubMed) and Scopus) were searched to identify articles. The keywords and search string used are detailed in Appendix A. In the first phase, articles were reviewed independently in increasing specificity via the title, abstract, and finally, via full text by NY and HAM. In the second phase, bibliographic references and citations of studies selected in phase one were extracted from Scopus and hand-searched for additional eligible studies based on the assumption that relevant studies cited others or were cited by other related studies. No publication date or publication status restriction was imposed. Discrepancies in the results of the selection were deliberated in team meetings. Where more than one report of a study existed, reports with a complete result were included. Where an institution published multiple reports from the same patient cohort but with different endpoints, all reports were included. Reports which combined head and neck carcinoma without providing separate analysis for oropharyngeal carcinoma were not included. Study search and selection were completed on 15 February 2023. The authors set notification alerts from Google Scholar for the selected articles, which notifies us of new citations. We believe this is required to allow a more dynamic inclusion of studies, as there is a time-lapse between the search and publication of the analysis.

### 2.3. Quality Assessment

We used an assessment tool from the National Heart, Lung, and Blood Institute—Quality Assessment Tool for Observational Cohort and Cross-Sectional Studies—to evaluate the quality of studies [29,30,31].

### 2.4. Data Review, Extraction, and Synthesis

Upon finalization of article selection, data extraction was performed by NY and was independently checked by HAM. Information was extracted into spreadsheets and included details of the articles, patients, proton therapy dose regime and technique, clinical factors, and demography. We also extracted the details of the instruments and endpoints used. If comparisons were made to photon-based therapy, the treatment and patient characteristics for photon therapy were also extracted. The corresponding authors were contacted to clarify missing/not reported information. These data were tabulated, separating the study characteristics, instruments and effect of technique, and clinical and dose factors into separate tables. Synthesis was performed based on the subdivisions of timeframes due to the known effect of time on the QOL and PRO outcomes.

### 2.5. Meta-Analyses

Due to the small number of eligible studies and studies from the same pool of patients reported by the same groups in multiple articles for different endpoints, meta-analysis is not warranted.

## 3. Results

### 3.1. Study Selection and Quality Assessment

The database queries produced 107 and 225 records from PubMed and Scopus, respectively (Figure 1, Appendix A). After removing duplicates, 252 reports were available, and 98 reviews, letters, and book chapters were removed, leaving 154 for title, abstract, and eligibility review for inclusion. Finally, seven met the inclusion criteria [32,33,34,35,36,37,38]. In the second phase, where citations of the previously selected reports were reviewed using Scopus, a source-neutral abstract and citation database, 240 articles were reviewed, and no additional papers were found. We followed the citations for the selected papers, and no additional papers were included. The included studies were found to be of good quality, with patients accrued in a single center in each study. This is not unexpected due to the limited number of proton centers to allow multicenter studies. Biases due to the lack of randomization include samples from temporally different cohorts, differences in access (insurance approval, socioeconomic status, distance to proton therapy center), and implicit or model-based patient selection to undergo PT (Appendix A). Sample size and calculations were rarely mentioned as patients were accrued chronologically.

### 3.2. Characteristics of Included Studies

Table 1 summarizes the characteristics of the selected studies, including 362 patients treated with proton and 497 patients treated with photon therapy. All studies reported prospectively collected outcome measurements. The publication dates ranged from 2016 to 2021, reflecting the recency of PT introduction into widespread practice and the emphasis on PRO and QOL measures.

### 3.3. QOL and PRO Measurements

The QOL and PRO instruments used were diverse (Table 2). Only two studies [33,34] utilized standard instruments, EORTC QLQ-H&N35 [39].

### 3.4. Studies Comparing Proton Therapy and Photon Radiotherapy

Five studies presented the comparison between proton therapy and proton radiotherapy in terms of QOL and PROs (Table 3). Blanchard et al. 2016 [32], Sio et al. 2016 [35], and Cao et al. 2021 [38] compared the outcomes of proton therapy to patients treated with IMRT, while more recent reports used patients treated with VMAT as a comparison. Randomization was not performed in any of the studies. Blanchard et al. performed 2:1 matching for laterality, site, HPV status, T and N status, smoking and chemotherapy, while other studies compared the clinical characteristics of patients treated with PBT and photons to detect demographic differences (Table 1). Patients treated with photon therapy were extracted from the historical cohort. Overall, in all QOL and PRO measures where the differences were significant, patients treated with proton therapy reported better outcomes, including lesser xerostomia, lesser cough, lesser need for nutritional supplements, lesser dysgeusia, better food taste, better appetite, less mucous and better general symptoms. However, in some sub-analyses performed by Manzar et al. for sexual symptoms, patients treated with IMRT performed better. In a significant number of endpoints and time points, proton- and photon-based therapy QOL and PRO outcomes did not significantly differ, and improved outcomes from PBT compared to photons were not universal in the included studies.

### 3.5. Effect of Time

In this review, we divided the time from treatment into several categories (Table 4): acute, subacute, late at <1 year, and late at ≥1 year. Compared to patients treated with photon radiotherapy, patients treated with proton beam therapy have better outcomes across most time points. Sio et al. is the only report which included the outcome during treatment and found no difference between the two treatment modalities. Blanchard et al. found a difference in xerostomia between PT and photon-based during subacute time points only, and no difference was found during the late time points. In contrast, Cao et al. found significant differences at later time points (18–24 months and 24–36 months) but no difference before that compared to photon-based therapy. Bagley et al. and Grant et al. compared the symptoms to baseline, and they found a similar pattern for xerostomia and dysphagia, where the worst happened during treatment, which improved during subacute and late phases. Scores, however, did not return to baseline. Focusing on xerostomia (Figure 2), the advantages of PBT were consistent across time points, and studies with PBT either produced better outcomes or had no significant difference to photon-based treatments.

### 3.6. Effect of Dose Factors

Only two studies analyzed the dose–outcome association, which found significant associations between oral cavity dose and xerostomia [36,38] (Table 3). The difference between doses received by patients treated with proton therapy and photon radiotherapy was a subject of studies by Sharma et al. and Cao et al., which found dose differences at the ipsilateral parotid, contralateral parotid, ipsilateral sublingual, contralateral sublingual, ipsilateral buccal, contralateral buccal, hard palate, tongue, upper lip, lower lip, and oral cavity (all favor PT) which complement findings from other dose comparison studies for head and neck cancers [14,33,34,43].

### 3.7. Effect of Clinical Factors

Bagley et al. found a significant positive correlation between the impact of time, baseline xerostomia score, stage, and N stages to the endpoints. The effect of the T stages was also reported by Grant et al. [37].

## 4. Discussion

We conducted a systematic review to methodically accumulate and synthesize the evidence regarding PROs and QOL following proton therapy for oropharyngeal carcinoma. This is an improvement from a systematic review by Verma et al., which examined QOL and PRO outcomes from patients with a variety of diagnoses treated with proton therapy, providing a good breadth of the issue but not depth [27]. Furthermore, we found six new articles that fulfilled our inclusion criteria compared to only one described by Verma et al. in 2017 [27]. This is expected, given the exponential growth of reports due to the recent availability of proton therapy in many centers. Based on this systematic review, we found the following: (1) studies frequently demonstrated the advantages of proton therapy compared to photon therapy, but biases due to the non-randomized nature of the studies limit the strength of this conclusion; (2) studies showed a significant decline in QOL and PROs following proton therapy at the acute stage, which improves over time but (3) does not revert to pretreatment levels; and (4) clinical, and dose features were infrequently studied.

Seven reports were included in this systematic review. However, five came from a single center: MD Anderson Cancer Center in Houston, Texas, USA. Furthermore, all centers reporting the outcomes were from the United States of America despite head and neck cancers being the second most frequently treated using proton therapy in Europe [44]. Only 107 proton therapy centers are operational worldwide, of which 43 are located in the USA (ptcog.ch). To put this into perspective, more than 14,000 high-energy photon therapy machines are available across the globe. As 38 new centers are being constructed, more reports on treatment outcomes following proton therapy may come soon. Second, centers are likely to focus on provider-reported outcomes for toxicity following treatments of oropharyngeal carcinoma, which are reported in several studies [45,46]. As the emphasis on PROs and QOL measures has been increasing in recent years due to a paradigm change to increase patients’ involvement in decision-making and capture outcomes that matter to patients, we are hopeful we will see reports with larger cohorts from other centers across the globe.

Generally, the PRO and QOL measures favor proton therapy. The results align with studies utilizing provider-rated toxicity measures [46,47]. Better outcomes for PBT may be explained by the improved normal tissue dose distributions achieved with proton therapy due to the beam characteristics. Several dose comparison studies have observed the superiority of proton therapy in terms of dose distribution compared to photon therapy [14,43,48]. The dose to the structures commonly associated with increased risks of xerostomia and dysphagia, including salivary glands and the larynx, was higher when VMAT or IMRT were utilized [48]. However, defining the temporal distribution when the differences were significant is challenging. Differences in xerostomia favoring PT during the first 3 months post-treatment lost significance after the acute phase in one study [32], while a number of domains remained significantly worse for photon therapy beyond the acute phase in two other studies [34,35]. In contrast, one further study. Cao et al. found that differences between PT and photons were only significant after 18 months. At least three hypotheses can be made from this observation. First, delayed recovery is associated with a low dose to the contralateral parotid gland which may explain the lack of PRO differences between PBT and photon-based therapies in the later follow-ups [49]. Second, after the initial stage of QOL following cancer treatment, patients may have adapted to the new norms and a more stoic approach towards life after cancer treatment, thus, reporting less bother [50]. Furthermore, effective symptom alleviation for some side effects, including saliva substitute for xerostomia, is available, reducing the bother [51]. Third, it should be noted that none of these studies randomized patients into proton or photon treatment arms. Consequently, bias cannot be discounted due to systematic differences between patients treated with proton and photon therapy. The lack of randomization may also impact future cost-effectiveness analysis [52]. Fortunately, an ongoing phase III randomized clinical trial, Toxicity Reduction Using Proton Beam Therapy for Oropharyngeal Cancer (TORPEDO), including extensive patient-reported outcome measures, will shed more light on the differences between IMPT and IMRT for oropharyngeal cancers in terms of QOL and patient-reported outcomes [53].

The main drawback of proton therapy is the high cost of constructing a cyclotron and maintaining the facility. To put the comparison in perspective, proton therapy costs approximately 2.4 times more than photon-based therapy. Frequently, the cost-effectiveness of proton therapy is being questioned [54]. Cost-effectiveness analysis showed that IMPT was only cost-effective for a fraction of younger patients with a high risk of profound reduction in long-term morbidity [54]. To identify patients expected to benefit from proton therapy compared to IMRT, selection based on normal tissue complication probability (NTCP) models has been implemented [55]. In this strategy, planning comparisons between photon and proton therapy were performed on an individual level. The reduction in multivariable, which numerically describes the relationship between the dose delivered to organs at risk and clinical factors and the predicted risk of radiation-induced side effects, was then employed to detect whether PT is clinically advantageous. With this strategy, resources can be optimally utilized to benefit patients without straining the healthcare system. This NTCP-based method, however, is subject to model- and dose-related uncertainties [56]. Furthermore, the high cost can be financially problematic for countries with limited resources, further widening the health disparities gap between countries [57,58,59], especially for treatments such as those for oropharyngeal carcinoma where it is expected that only 0 to 0.4% probability that proton therapy was cost-effective for 65- and 55-year-old patients [54], thus its utilization in low- and middle-income countries is likely to be societally unacceptable and should receive lower priority than pediatric cancers [60]. Fortunately, the cost of proton therapy is continuously decreasing with more compact single rooms with small cyclotrons, dielectric wall accelerators (DWAs), and laser-driven systems, allowing space-related costs to be further minimized [61]. Furthermore, the treatment delivery is also improving with the introduction of multifield optimization intensity-modulated proton therapy, significantly reducing integral doses [16]. Proton arc therapy can improve the treatment delivery efficiency and streamline the treatment workflow [62].

We should note some limitations of the systematic review. First, there were only a small number of studies conducted in a limited number of centers with the potential overlap between cohorts, and their non-randomized nature may limit the applicability of this review. Furthermore, inconsistency between included studies in terms of tools used while measuring the QOL, target dose, and combination with chemotherapy may limit the generalizability of the outcomes of this study. Second, no meta-analyses can be performed due to the lack of independence across studies with reports focusing on different endpoints in a group of similar patients. Third, it is acknowledged that selection bias, where negative studies are less likely to be published and thus not be searchable, may influence the observation.

## 5. Conclusions

Proton therapy may improve patient-reported and quality-of-life outcomes for patients treated for oropharyngeal carcinoma, especially up to one year post-treatment. Symptoms may improve over time but may not return to baseline. The current evidence is, however, limited. As proton therapy becomes more widely available, it is hoped that further outcome data will allow more definitive conclusions to be reached. The limitation highlighted in the present study calls for collaboration between proton therapy centers to make multicenter randomized studies, including carefully selected patients and more standardized treatment applications.

## Figures and Tables

**Figure 1 cancers-15-02252-f001:**
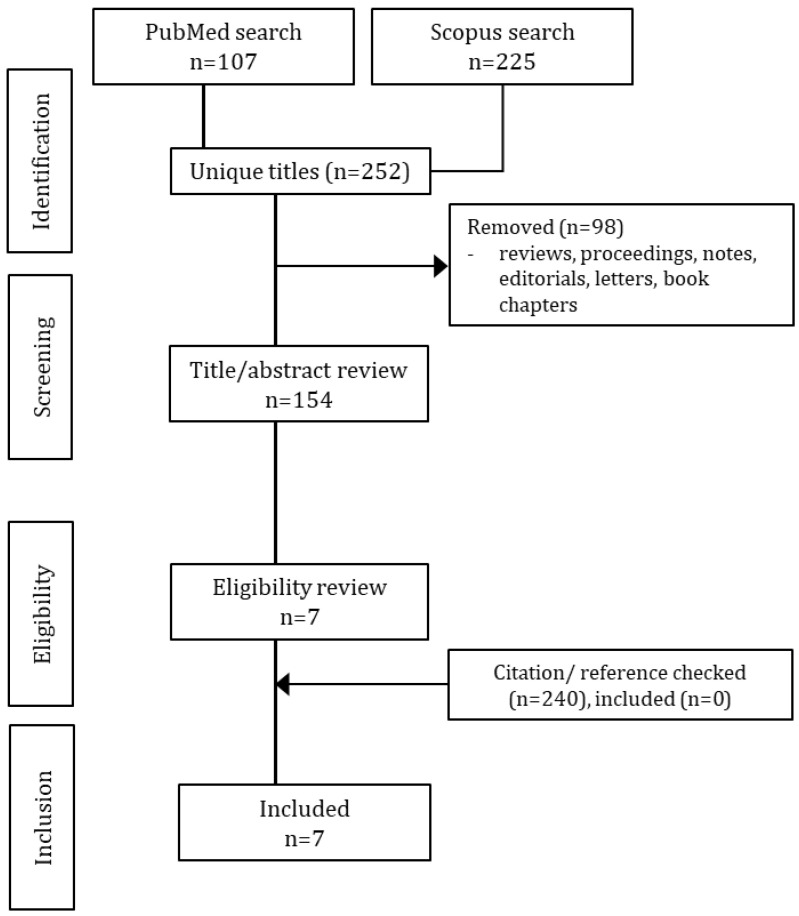
Identification of articles for inclusion. Eligibility was determined using PICOS criteria (Appendix A).

**Figure 2 cancers-15-02252-f002:**
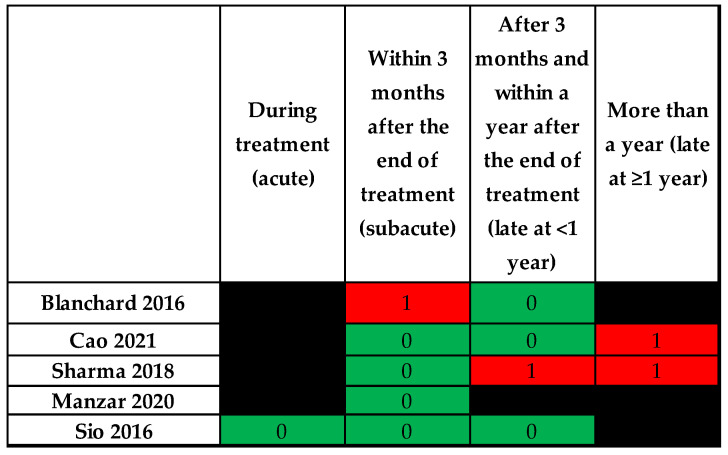
Patient-reported xerostomia between PBT and photon-based therapy at different time points. Red represents a significant difference (favoring PBT), and green represents no significant difference. None of the time points favor photon-based therapy (Blanchard 2016 [32], Cao 2021 [38], Sharma 2018 [34], Manzar 2020 [33] and Sio 2016 [35]).

**Table 1 cancers-15-02252-t001:** Study characteristics.

Reference	No.	Stage III/IV (%)	Female (%)	Median Age (Range)	% HPV Positive (+), Unknown (?)	Type of Proton Therapy	Dose (Gy RBE)	Other Treatments	Therapy Comparison	Details of Therapy Comparison
Bagley 2020 [36]	69	AJCC7 stage III–IV, M0—100	13	64 (37–84)	+84 ? 14	Spot-scanning IMPT	Median—69.3, range 60–70	Induction—5, concurrent—38, Induction + concurrent—11	-	-
Blanchard 2016 [32]	50	T3–T4—20 N2–N3—80	14.7	IMRT—55.5 (34–78), IMPT—61 (37–84)	+88 ? 10	Spot-scanning IMPT	Small volume disease—66, advanced disease—70	Concurrent—64%	100 IMRT	2:1, matched laterality, site, HPV, T and N status, smoking, and chemotherapy
Cao 2021 [38]	103	T3–T4—IMRT (31.9), IMPT (35); N2–N3—IMRT (82.3), IMPT (77.1)	IMRT—14.2, IMPT—12.6	IMRT—59 (32–84), IMPT—60 (33–85)	IMRT (+68.8 ? 18.1) IMPT (+76.7 ? 17.4)	spot-scanning IMPT	With concurrent chemo—70 Gy Without chemo—60 Gy	Neoadjuvant—35%, Concurrent—69%	429 IMRT	No significant difference for demographic and treatment factors tested (age, sex, race, tumor site, location, clinical stage, human papillomavirus status, or chemotherapy received)
Grant 2020 [37]	71	AJCC 7th edition stage III/IV—100	12.7	63 (37–84)	+85.9 ? 7.0	IMPT	Range 66–70 Gy	Induction—5, concurrent—41, induction + concurrent—10	-	-
Manzar 2020 [33]	46	AJCC 7 th edition Stage III/IV—84.8	12.5	VMAT—61 (42–88), IMPT—66 (40–79)	+76.1 ? 13.0	spot-scanning IMPT	Adjuvant, range 60–66; definitive70	Concurrent—36	259 VMAT	Significant difference: age (IMPT older) smoking status and pack-years (VMAT higher), dose category (more definitive RT in IMPT)
Sharma 2018 [34]	31	Stage I–III—13 IVA 87	VMAT—18 Proton—13	VMAT (mean)—58, Proton (mean)—60	Not stated	spot-scanning, single-field uniform dose	Median 61.7	Chemotherapy—12	33 VMAT	No significant difference
Sio 2016 [35]	35	Stage III–IV—94.3	IMRT—8.7, IMPT—14.3	IMRT (mean, SD)—58.2 (9.9) IMPT (mean, SD)—59.1 (10.2)	+74.3 ? 20	Spot-scanning IMPT	Median 70.0, range 59.0–70.0	Concurrent chemotherapy—all, induction—26	46 IMRT	Significant difference: location (more tonsil), T-stage (more T3–T4), lower induction chemotherapy, higher total radiation dose

**Table 2 cancers-15-02252-t002:** Quality of life and patient-reported outcomes measures and assessments.

Reference	QOL/PRO Measures	Endpoints	Type and Frequency of Assessment	Median Follow-Up
Bagley 2020 [36]	15-item Xerostomia-Related QoL Scale (XeQoLS); range score—0–75 [40]	Mean XeQoLS scores and subdomain (physical, personal, pain, social)	Prospective—baseline, 6 weeks on-treatment, and follow-up visits at 10 weeks and at 6, 12, and 24 months	64 weeks from the start of treatment
Blanchard 2016 [32]	Not specified; range 0–3 scale from none to severe	Grade 2–3 patient-rated fatigue and dry mouth	Prospective—during treatment, 3 months after, and 1 year after	29 months (range 8–49)—IMPT, 33 months (range 2–55)—IMRT
Cao 2021 [38]	Eight-item self-reported xerostomia-specific questionnaire; range score—0–100 [41]	Moderate–severe score ≥ 50 and no–mild score < 50	Prospective—every 3 months and clustered into 0–6, 6–9, 9–12, 12–18, 18–24, and 24–36 months	36.2 months
Grant 2020 [37]	MD Anderson Dysphagia Inventory; 20 questions from which global, composite, functional, emotional, and physical scores were derived and normalized; score range 20 (extremely low functioning) to 100 (high functioning)	Score changes over time	Prospective—baseline, treatment week 6, follow-up week 10, month 6, year 1, and year 2	More than 50% of patients evaluable at 24 months
Manzar 2020 [33]	EORTC QLQ-H&N35—35 questions; 35 questions covering aspects of QOL [39]	End-of-treatment scores for each question	Prospective—for QoL only end of treatment analyzed	12 months (IMPT) and 30 months (VMAT)
Sharma 2018 [34]	QLQ-30 version 3, EORTC QLQ-H&N35, and the Groningen Xerostomia, Work Status, and Performance Status Scale—Head and Neck Cancer (GRIX) questionnaires, normalized a 0 to 100 scale; EORTC—general health domain, physical and role function, overall xerostomia, dental issues, head and neck pain, and fatigue scores; GRIX—day and night xerostomia and separate subscales for sticky saliva [39,42]	Score at 3, 6, and 12 months	Prospective—pretreatment and at 3, 6, and 12 months	Not mentioned
Sio 2016 [35]	MD Anderson Symptom Inventory-Head and Neck Cancer (MDASI-HN)—top 11 most severe burdens	During treatment (acute phase), within the first 3 months after treatment (subacute phase), and afterward (chronic phase)	Prospective—weekly during the 6- to 7-week radiotherapy period (the acute phase). Data in the subacute phase were obtained during the first 3 months after the end of radiotherapy	7.7 (IQR 3.97–22.77) months—IMPT, 2.68 (0.30–10.27) months—IMRT

**Table 3 cancers-15-02252-t003:** Significant endpoints and significant factors impacting endpoints.

Reference	Endpoints	Statistically Significant Endpoints	Non-Statistically Significant Endpoints	Significant Clinical Factors	Dose Factors
*Comparison to Baseline*					
Bagley 2020 [36]	mean XeQoLS scores and subdomain (physical, personal, pain, social)	General xerostomia, including physical, personal, pain, and social domains Baseline: 0.24 ± 0.57 6 weeks: 2.00 ± 1.01 10 weeks: 1.03 ± 0.76 6 months: 0.97 ± 0.78 1 year: 0.82 ± 0.69 2 years: 0.70 ± 0.75 (all *p* < 0.001)	-	Time, baseline XeQoLS score, stage and N status	Univariate—oral cavity dose Multivariate—not significant
Grant 2020 [37]	MDADI score changes over time	Poor composite score for dysphagia Baseline: 5.6% 6 weeks: 61.2% 10 weeks: 19.1% 6 months: 13.3% 1 year: 13.5% 2 years: 11.1%	-	T-stage	Not studied
*Comparison to photon-based therapy*					
Blanchard 2016 [32]	Grade 2–3 patient-rated fatigue and dry mouth	Xerostomia at 3 months (favors IMPT 42% vs. 61.2%, *p* = 0.009)	Fatigue at 3 months (IMPT = 40.8% vs. IMRT = 36.2%); fatigue (IMPT = 14.6% vs. IMRT = 22.1%) and xerostomia (IMPT = 42% vs. IMRT = 47.2%) at 1 year	Not studied	Not studied
Cao 2021 [38]	Moderate–severe xerostomia score ≥ 50	Moderate–severe at 18–24 months (favors IMPT 6% vs. 20%; *p* = 0.025) and at 24–36 months (favors IMPT 6% vs. 20%; *p* = 0.01)	Up until 18 months after treatment 0–6 months (IMPT 38% vs. IMRT 37%), 6–9 months (IMPT 25% vs. IMRT 28%), 9–12 months (IMPT 10% vs. IMRT 16%), 12–18 months (IMPT 7% vs. IMRT 7%)	18–24 months—disease site (base of tongue vs. tonsil/other: OR = 0.320, *p* = 0.009) 24–36 months—gender (male vs. female, OR = 2.786, *p* = 0.023), concurrent CT (OR = 0.349, *p* = 0.024)	0–6 months—higher Dmean, V5, V10, V15, V20, V25, V30, V35, V40, and V45 of the contralateral parotid gland for moderate–severe 24–36 months—higher V25, V30, V35, V40, V45, V50, V55, V60, V65, and V70 of the oral cavity
Manzar 2020 [33]	End-of-treatment scores for each question	Overall (mean difference to baseline): Cough (IMPT 6.7 vs. IMRT 29.3, *p* = 0.003), need for nutritional supplements (IMPT 26.5 vs. IMRT 48.1, *p* = 0.007), and dysgeusia (IMPT 3.7 vs. IMRT 6.9, *p* = 0.043) (all favor IMPT)	EORTC H&N QLQ-35 questions not stated in the previous cell	Not studied	Significant dose difference between PT and VMAT. Associations to endpoints not analyzed
Sharma 2018 [34]	Score at 3, 6, and 12 months.	3 months—dental problem (IMPT 0% vs. IMRT 19.05%, *p* = 0.016) 6 months—moderate to severe dry mouth (IMPT 22.22% vs. IMRT 63.16%, *p* = 0.02), xerostomia day (IMPT 25.80% vs. IMRT 39.20%, *p* = 0.038), xerostomia night (IMPT 22.80% vs. IMRT 35.10%, *p* = 0.042), dental problems (IMPT 1.96% vs. IMRT 17.54%, *p* = 0.048), physical function (IMPT 97.04% vs. IMRT 89.47%, *p* = 0.006), role function (IMPT 96.30% vs. IMRT 76.32%, *p* = 0.0008) 12 months—H&N pain (IMPT 8.33% vs. IMRT 21.97%, *p* = 0.011), xerostomia (IMPT 23.53% vs. IMRT 54.55%, *p* = 0.003), moderate-severe dry mouth (IMPT 11.76% vs. IMRT 50.00%, *p* = 0.038), role function (IMPT 81.86% vs. IMRT 72.73%, *p* = 0.041)	Fatigue, sticky saliva (day general), and global health and time points not stated in the previous cell	Not studied	Significant dose difference between PT and IMRT (all favoring PT); associations to endpoints not analyzed
Sio 2016 [35]	Average symptom burden in the top 11 (most severe) items in the MDASI during treatment (acute phase) within the first 3 months after treatment (subacute phase), and afterward (chronic phase)	Burden in subacute phase—food taste (IMPT score (SD) 5.76 (3.60) vs. IMRT 7.70 (2.44), *p* = 0.010) and appetite (IMPT 4.68 (3.53) vs. IMRT 6.37 (3.21), *p* = 0.048). Burden in chronic phase—appetite (IMPT 2.12 (3.08) vs. IMRT 4.14 (3.01), *p* = 0.036) moderate to severe symptoms—subacute phase—food taste and mucus (favor IMPT, *p* < 0.039 for both) mean of top 5 MDASI higher during subacute phase for IMRT IMPT 8.7 (8.8) vs. IMRT 36.4 (22.4)	Other top 11 symptoms—Dry mouth, swallowing/chewing, fatigue, pain, sleep, mouth sores, drowsiness, and distress.	Not studied	Not studied

**Table 4 cancers-15-02252-t004:** Significant endpoints based on the time point divided into acute, subacute, late at <1 year, and late at ≥1 year.

	During Treatment (Acute)	Within 3 Months after the End of Treatment (Subacute)	After 3 Months and Within a Year after the End of Treatment (Late at <1 Year)	More than a Year (Late at ≥1 Year)
*Comparison to Photon*				
Blanchard 2016 [32]	-	Xerostomia score (favors IMPT), fatigue—no significant difference	Xerostomia and fatigue score—no significant difference	-
Cao 2021 [38]		Xerostomia—no significant difference	Xerostomia—no significant difference	Moderate–severe xerostomia (favors IMPT)
Sharma 2018 [34]	-	Dental problem (favors IMPT)	Moderate to severe dry mouth, xerostomia day, xerostomia night, dental problems, physical function, role function (all favor IMPT)	H&N pain, xerostomia, moderate–severe dry mouth, role function (all favor IMPT)
Manzar 2020 [33]	-	Cough, need for nutritional supplements and dysgeusia (favors IMPT)	-	-
Sio 2016 [35]	No difference.	Mean symptom scores—food taste and appetite (favors IMPT); moderate to severe symptoms—food taste and mucus (favors IMPT)	Mean symptom scores—appetite (favors IMPT)	-
*Comparison to baseline*				
Bagley 2020 [36]	Worst xerostomia score	Significantly better xerostomia score than acute, significantly worse than baseline	Significantly better xerostomia than subacute, significantly worse than baseline	Not significantly different xerostomia than at 1 year, significantly worse than baseline
Grant 2020 [37]	Worst dysphagia score	Significantly better dysphagia score than acute, significantly worse than baseline	Significantly better dysphagia score than subacute, significantly worse than baseline	Not significantly different dysphagia score than at 1 year, significantly worse than baseline

## Data Availability

Data are available in Appendix A. Further data are available from the corresponding author upon reasonable request.

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
