# Peer review of "Quality of Life and Patient-Reported Outcomes Following Proton Therapy for Oropharyngeal Carcinoma: A Systematic Review"

_cancers, 2023, doi:10.3390/cancers15082252_

Round 1
Reviewer 1 Report
Dear authors,
I have read and reviewed this manuscript with a great interest. In the attached document, you can find my suggestions and comments.
Best regards

Author Response
Thank you for your insightful comments and suggestions. We have thoroughly considered all and added replies to your comments in the pdf file.

Reviewer 2 Report
This is a nicely performed systematic review investigating QoL and PRO in patients treated with PBT for oropharyngeal cancers. The authors appear to have done a rigorous search and followed established guidelines for performing a systematic review. The review is limited (through no fault of the authors) by the fact that there are a very limited number of studies available, none of which are randomised. I'm therefore not convinced the outcomes are of sufficient impact to warrant publication in this journal. Some of the written English could be improved and at times this is a distraction from a fuller review of the content. I was unable to download the supplementary files as the file type was not recognised. I'm happy to review these in a different format.
The authors need to make clearer in the results section (3.4) that none of the comparative studies were randomised (this is a big weakness in the existing evidence) as at present this is not clear. How were the photons controls therefore selected? ?historical controls? Please expand a little on this. Also, in section 3.4 it needs to be clearer that the finding of improved outcomes from PBT compared to photons was not universal in the included studies. Specify the outcomes that Manzar demonstrated to favour photons here too.
In Table 3, remove the 'the' from the table title. Maybe add a column to make it clear what the comparison is, i.e. change over time or comparisons with photons. When there hasn't been a comparison to photons, make it clear what the direction of the significant finding is, for example, does composite dysphagia score worsen of improve over time?
Section 3.5- for the sentences beginning Blanchard et al and Coa et al, make it clear that you are talking about comparisons with photons at each time point, whereas for Bagley and Grant, this is PBT only and compared to baseline symptoms.
Section 3.6- I find this hard to follow.... I think the sentence beginning The difference between doses...' provides the same information as the next sentence, beginning 'Doses to multiple structures...'. Please make it clearer if these are meant to be delivering a different piece of information. The English makes it hard to follow (see below).
Section 3.7. Again, I think the first and second sentences are trying to say the same thing and the English compromises understanding of what this is.
In the discussion, I'm not sure the three hypotheses to account for the differing results between studies in terms of temporal changes in QOL/ PRO really explain these- please re-word and make clearer.
In the final paragraph of the discussion, re limitations, I think you could add that studies were often had relatively low patient numbers, and re-iterate there may be overlap between cohorts. And mention again about non-randomised nature.
The reference numbers should be provided in all the tables and throughout the results to make it easier for the reader (i.e. these are missing from the tables and sections 3.4, 3.5, some of 3.6 and 3.7).
Specifically:
Abstract: change 'good' option to 'potentially attractive' option,
remove 'changes' from the next sentence (and throughout the manuscript- the word 'changes' is not necessary and not always relevant,
in the results, 'Five compared PT and photon-based therapy. None of them is a randomised controlled trial'.. change to Five compared PT and photon-based therapy, although none were randomised controlled trials'
'The PRO and QOL change following PT improves but never to the baseline level'.
.. change to: PRO and QOL improve following PT but do not appear to return to baseline.
'Biases due to the non-randomised study design remain obstacles to a concrete conclusion'... change 'concrete' to 'firm'
'Whether or not it is cost-effective should be the subject of further investigation'... change to 'Whether or not PT is cost-effective should be the subject of further investigation'
Intro
'As there are multitudes of studies that have empirically demonstrated the impact of dose distributions to salivary glands [5-7], swallowing muscles [8, 9], oral cavity [3] and other normal tissues to patient-reported outcomes (PRO) and QOL measures in photon based therapy'.. change to There are numerous studies that have empirically demonstrated the impact of photon dose to salivary glands [5-7], swallowing muscles [8, 9], oral cavity [3] and other normal tissues on patient-reported outcomes (PRO) and QOL measures'
'Several dose comparison studies have shown significant dose reduction favouring PT [11, 12].'.. change to 'Several dose comparison studies have shown significant normal tissue dose reductions favouring PT [11, 12]'
'Deviating the given dose distribution from the intended distribution due to uncertainties from intrafraction motion and patient set-up error is significantly more prominent in the pencil beam scanning technique'.... change to 'Deviations of the given dose distribution from the intended distribution as a result of uncertainties arising form from intrafraction motion and patient set-up error are significantly more prominent with pencil beam scanning techniques'.
'However, as more centres offer PT and increasing interest in measuring outcomes that are more patient-centred and connect to patients more meaningfully, we may see more clinical results especially involving PRO and QOL mature.'..... change to 'However, as more centres offer PT and as there is increasing interest in measuring outcomes that are more patient-centred and connect to patients more meaningfully, we may see more clinical results that involving PRO and QOL.'
'In this study, we aimed to evaluate the evidence of the QOL and PRO changes following PT of OC with a secondary aim to compare the outcomes of PT and photon-based therapy'.... change to 'In this study, we aimed to evaluate the evidence around QOL and PRO measures following PT of OC, with a secondary aim to compare the outcomes of PT and photon-based therapy'
Results
'The included studies were found to be of good quality with patients accrued in a centre which is expected due to the limited number of proton centres'... I don't really understand this.... do you mean: 'The included studies were found to be of good quality, with patients accrued in a single centre in each study, which is not unexpected due to the limited number of proton centres'?
'In this review, we divided the time into several time points (Table 4); acute, subacute, late at <1 year and late at ≥1 year.'.. change to: 'In this review, we divided the time from treatment into several categories (Table 4); acute, subacute, late at <1 year and late at ≥1 year.
'The scores, however, remained worse than the baseline'.... change to: 'Scores, however, did not return to baseline'
''The difference between doses received by patients treated with proton therapy and photon radiotherapy was a subject of studies by Sharma et al. and Cao et al., which found dose differences at ipsilateral parotid, contralateral parotid, ipsilateral sublingual, contralateral sublingual, ipsilateral buccal, contralateral buccal, hard palate, tongue, upper lip, lower lip, oral cavity. Doses to multiple structures were found to receive a lower radiation dose in proton therapy compared to photon therapy reported by Sharma et al., Manzar et al. and Cao et al., which complement findings from other dose comparison studies for head and neck cancers [13, 23, 24, 30]'.....I think both these sentences are saying the same thing essentially? I'd remove this first and say; 'Multiple normal structures were found to receive lower radiation doses with PT compared to photon therapy (Sharma et al., Manzar et al. and Cao et al.), which complement findings from other dose comparison studies for head and neck cancers [13, 23, 24, 30].' If this is what you are meaning.
'Bagley et al. found a significant impact of time, baseline xerostomia score, stage and N. The T and N stages, baseline status and time from the radiotherapy were found to be significant predictors for PRO and QOL.'.. this is very hard to understand.... do you mean: Bagley et al. found that T and N stages, baseline status and time from the radiotherapy were significant predictors of PRO and QOL.'?
Discussion
'We conducted a systematic review to methodically accumulate and synthesise the evidence of PRO and QOL changes following proton therapy of oropharyngeal carcinoma'... change to: 'We conducted a systematic review to methodically accumulate and synthesise the evidence regarding PRO and QOL following proton therapy for oropharyngeal carcinoma'
'This is an improvement from a systematic review by Verma et al., which combined treatments of many diagnoses treated with proton therapy, providing a good breadth of the issue but not depth [17]'.... change to: 'This is an improvement from a systematic review by Verma et al., which examined QOL and PRO outcomes from patients with a variety of diagnoses treated with proton therapy, providing a good breadth of the issue but not depth [17]'
'Based on this systematic review, we found; 1) studies showed the advantages of proton therapy compared to photon therapy; however, biases due to the non-randomised nature of the studies limit the confidence for a firm conclusion, 2) studies showed a significant decline in functions following proton therapy at the acute stage which improves and 3) functions do not revert to pretreatment status'.... change to: 'Based on this systematic review, we found; 1) studies frequently demonstrated advantages of proton therapy compared to photon therapy; however, biases due to the non-randomised nature of the studies limit the strength of this conclusion, 2) studies showed a significant decline in QOL and PRO following proton therapy at the acute stage which improves over time but 3) do not revert to pretreatment levels.'
'...we are optimistic to see reports with larger cohorts...'... change to '...we are hopeful we will see reports from larger cohorts...'
'This can be simply explained by the better dose distribution using proton therapy due to the beam characteristics, as several dose comparison studies have observed the superiority of proton therapy as compared to photon'... change to: 'This maybe explained by the improved normal tissue dose distributions achieved with proton therapy due to the beam characteristics, as several dose comparison studies have observed the superiority of proton therapy as compared to photon therapy'
'However, defining the temporal distribution when the differences were significant is challenging. The change for xerostomia was found to be no longer significant after the acute phase [22], and the number of domains remained significantly worse for photon therapy after the acute phase was also reduced [24, 25]. In contrast, Cao et al. found the opposite, where the differences were only significant after 18 months'... change to: 'However, defining the temporal distribution when the differences were significant is challenging. Differences in xerostomia favouring PT during the first 3 months post treatment lost significance after the acute phase in one study [22], while a number of domains remained significantly worse for photon therapy beyond the acute phase in two other studies [24, 25]. In contrast, one further study. Cao et al. found differences between PT and photons were only significant after 18 months'
'First, the higher dose received from photon therapy...'.. change to 'First, the higher normal tissue doses received from photon therapy....'.
'First, a small number of the available study were conducted in a limited number of centres may limit the applicability of this review'..... change to 'First, there were only a small number of studies, which were conducted in a limited number of centres. This may limit the applicability of this review'
Conclusion
'Proton therapy may improve patient-reported and quality-of-life outcomes for patients treated for oropharyngeal carcinoma, especially in acute and early late time points, which improve in later time points. This observation can be proven or challenged as more data available as proton therapy becomes more widely available.'.. change to: 'Proton therapy may improve patient-reported and quality-of-life outcomes for patients treated for oropharyngeal carcinoma, especially up to one year post-treatment. Symptoms may improve over time but may not return to baseline. The current evidence is, however, limited. As proton therapy becomes more widely available, it is hoped that further outcome data allow more definitive conclusions to be reached.'
Author Response
This is a nicely performed systematic review investigating QoL and PRO in patients treated with PBT for oropharyngeal cancers. The authors appear to have done a rigorous search and followed established guidelines for performing a systematic review. The review is limited (through no fault of the authors) by the fact that there are a very limited number of studies available, none of which are randomised. I'm therefore not convinced the outcomes are of sufficient impact to warrant publication in this journal. Some of the written English could be improved and at times this is a distraction from a fuller review of the content. I was unable to download the supplementary files as the file type was not recognised. I'm happy to review these in a different format.
Thank you for your comments and suggestions. We have checked the grammar. The supplementary materials have been reuploaded.
The authors need to make clearer in the results section (3.4) that none of the comparative studies were randomised (this is a big weakness in the existing evidence) as at present this is not clear. How were the photons controls therefore selected? ?historical controls? Please expand a little on this. Also, in section 3.4 it needs to be clearer that the finding of improved outcomes from PBT compared to photons was not universal in the included studies. Specify the outcomes that Manzar demonstrated to favour photons here too.
The patients were selected from the same centre as comparison. We have also highlighted the non-randomised nature of the included studies in this section. We have added the sentence highlighting that the effects were not universal. The outcome which was significantly worse for photon-based RT from Manzar et al. has been added into the text.
In Table 3, remove the 'the' from the table title. Maybe add a column to make it clear what the comparison is, i.e. change over time or comparisons with photons. When there hasn't been a comparison to photons, make it clear what the direction of the significant finding is, for example, does composite dysphagia score worsen of improve over time?
We have divided the table into the type of comparisons. We also add the direction of association.
Section 3.5- for the sentences beginning Blanchard et al and Coa et al, make it clear that you are talking about comparisons with photons at each time point, whereas for Bagley and Grant, this is PBT only and compared to baseline symptoms.
Thank you for highlighting this. We agreed with this observation and have edited the sentence accordingly.
Section 3.6- I find this hard to follow.... I think the sentence beginning The difference between doses...' provides the same information as the next sentence, beginning 'Doses to multiple structures...'. Please make it clearer if these are meant to be delivering a different piece of information. The English makes it hard to follow (see below).
Thank you for your comment. We have merged the sentences and parts where they are repetitive were removed. It reads:
The difference between doses received by patients treated with proton therapy and photon radiotherapy was a subject of studies by Sharma et al. and Cao et al., which found dose differences at ipsilateral parotid, contralateral parotid, ipsilateral sublingual, contralateral sublingual, ipsilateral buccal, contralateral buccal, hard palate, tongue, upper lip, lower lip, oral cavity which complement findings from other dose comparison studies for head and neck cancers [13, 25, 26, 35].
Section 3.7. Again, I think the first and second sentences are trying to say the same thing and the English compromises understanding of what this is.
The second sentence has been edited. It reads:
Bagley et al. found a significant positive correlation of the impact of time, baseline xe-rostomia score, stage and N stages to the endpoints. The effect of the T stages was reported by Grant et al. [29].
In the discussion, I'm not sure the three hypotheses to account for the differing results between studies in terms of temporal changes in QOL/ PRO really explain these- please re-word and make clearer.
We have reworded the hypotheses and added references.
In the final paragraph of the discussion, re limitations, I think you could add that studies were often had relatively low patient numbers, and re-iterate there may be overlap between cohorts. And mention again about non-randomised nature.
We have combined the comments from all reviewers and added the limitations as suggested. The paragraph now reads:
We should note some limitations of the systematic review. First, a small number of the available study were conducted in a limited number of centres with potential overlap between cohorts and their nonrandomised nature may limit the applicability of this re-view. Furthermore, inconsistency between included studies in terms of tools used while measuring the QOL, target dose and combination with chemotherapy may limit the gen-eralisability of the outcomes of this study. Second, no meta-analyses can be performed due to the lack of independence across studies. Third, it is acknowledged that selection bias, where negative studies are less likely to be published and thus not be searchable, may in-fluence the observation.
The reference numbers should be provided in all the tables and throughout the results to make it easier for the reader (i.e. these are missing from the tables and sections 3.4, 3.5, some of 3.6 and 3.7).
Added.
Specifically:
We would like to thank you for the detailed read of the manuscript.
Abstract: change 'good' option to 'potentially attractive' option,
Done.
remove 'changes' from the next sentence (and throughout the manuscript- the word 'changes' is not necessary and not always relevant,
Thank you for your observation. The word “changes” have been removed.
in the results, 'Five compared PT and photon-based therapy. None of them is a randomised controlled trial'.. change to Five compared PT and photon-based therapy, although none were randomised controlled trials'
Done
'The PRO and QOL change following PT improves but never to the baseline level'.
.. change to: PRO and QOL improve following PT but do not appear to return to baseline.
Done
'Biases due to the non-randomised study design remain obstacles to a concrete conclusion'... change 'concrete' to 'firm'
Done
'Whether or not it is cost-effective should be the subject of further investigation'... change to 'Whether or not PT is cost-effective should be the subject of further investigation'
Done.
Intro
'As there are multitudes of studies that have empirically demonstrated the impact of dose distributions to salivary glands [5-7], swallowing muscles [8, 9], oral cavity [3] and other normal tissues to patient-reported outcomes (PRO) and QOL measures in photon based therapy'.. change to There are numerous studies that have empirically demonstrated the impact of photon dose to salivary glands [5-7], swallowing muscles [8, 9], oral cavity [3] and other normal tissues on patient-reported outcomes (PRO) and QOL measures'
'Several dose comparison studies have shown significant dose reduction favouring PT [11, 12].'.. change to 'Several dose comparison studies have shown significant normal tissue dose reductions favouring PT [11, 12]'
'Deviating the given dose distribution from the intended distribution due to uncertainties from intrafraction motion and patient set-up error is significantly more prominent in the pencil beam scanning technique'.... change to 'Deviations of the given dose distribution from the intended distribution as a result of uncertainties arising form from intrafraction motion and patient set-up error are significantly more prominent with pencil beam scanning techniques'.
'However, as more centres offer PT and increasing interest in measuring outcomes that are more patient-centred and connect to patients more meaningfully, we may see more clinical results especially involving PRO and QOL mature.'..... change to 'However, as more centres offer PT and as there is increasing interest in measuring outcomes that are more patient-centred and connect to patients more meaningfully, we may see more clinical results that involving PRO and QOL.'
'In this study, we aimed to evaluate the evidence of the QOL and PRO changes following PT of OC with a secondary aim to compare the outcomes of PT and photon-based therapy'.... change to 'In this study, we aimed to evaluate the evidence around QOL and PRO measures following PT of OC, with a secondary aim to compare the outcomes of PT and photon-based therapy'
Results
'The included studies were found to be of good quality with patients accrued in a centre which is expected due to the limited number of proton centres'... I don't really understand this.... do you mean: 'The included studies were found to be of good quality, with patients accrued in a single centre in each study, which is not unexpected due to the limited number of proton centres'?
You are right. We have edited the sentence with appropriate punctuations.
'In this review, we divided the time into several time points (Table 4); acute, subacute, late at <1 year and late at ≥1 year.'.. change to: 'In this review, we divided the time from treatment into several categories (Table 4); acute, subacute, late at <1 year and late at ≥1 year.
Done.
'The scores, however, remained worse than the baseline'.... change to: 'Scores, however, did not return to baseline'
Done.
''The difference between doses received by patients treated with proton therapy and photon radiotherapy was a subject of studies by Sharma et al. and Cao et al., which found dose differences at ipsilateral parotid, contralateral parotid, ipsilateral sublingual, contralateral sublingual, ipsilateral buccal, contralateral buccal, hard palate, tongue, upper lip, lower lip, oral cavity. Doses to multiple structures were found to receive a lower radiation dose in proton therapy compared to photon therapy reported by Sharma et al., Manzar et al. and Cao et al., which complement findings from other dose comparison studies for head and neck cancers [13, 23, 24, 30]'.....I think both these sentences are saying the same thing essentially? I'd remove this first and say; 'Multiple normal structures were found to receive lower radiation doses with PT compared to photon therapy (Sharma et al., Manzar et al. and Cao et al.), which complement findings from other dose comparison studies for head and neck cancers [13, 23, 24, 30].' If this is what you are meaning.
Apologies for this error. We have edited the sentences as suggested by all reviewers.
'Bagley et al. found a significant impact of time, baseline xerostomia score, stage and N. The T and N stages, baseline status and time from the radiotherapy were found to be significant predictors for PRO and QOL.'.. this is very hard to understand.... do you mean: Bagley et al. found that T and N stages, baseline status and time from the radiotherapy were significant predictors of PRO and QOL.'?
We have edited the sentences. It now reads:
Bagley et al. found a significant positive correlation of the impact of time, baseline xe-rostomia score, stage and N stages to the endpoints. The effect of the T stages was also re-ported by Grant et al. [29].
Discussion
'We conducted a systematic review to methodically accumulate and synthesise the evidence of PRO and QOL changes following proton therapy of oropharyngeal carcinoma'... change to: 'We conducted a systematic review to methodically accumulate and synthesise the evidence regarding PRO and QOL following proton therapy for oropharyngeal carcinoma'
'This is an improvement from a systematic review by Verma et al., which combined treatments of many diagnoses treated with proton therapy, providing a good breadth of the issue but not depth [17]'.... change to: 'This is an improvement from a systematic review by Verma et al., which examined QOL and PRO outcomes from patients with a variety of diagnoses treated with proton therapy, providing a good breadth of the issue but not depth [17]'
Done.
'Based on this systematic review, we found; 1) studies showed the advantages of proton therapy compared to photon therapy; however, biases due to the non-randomised nature of the studies limit the confidence for a firm conclusion, 2) studies showed a significant decline in functions following proton therapy at the acute stage which improves and 3) functions do not revert to pretreatment status'.... change to: 'Based on this systematic review, we found; 1) studies frequently demonstrated advantages of proton therapy compared to photon therapy; however, biases due to the non-randomised nature of the studies limit the strength of this conclusion, 2) studies showed a significant decline in QOL and PRO following proton therapy at the acute stage which improves over time but 3) do not revert to pretreatment levels.'
Done.
'...we are optimistic to see reports with larger cohorts...'... change to '...we are hopeful we will see reports from larger cohorts...'
Done.
'This can be simply explained by the better dose distribution using proton therapy due to the beam characteristics, as several dose comparison studies have observed the superiority of proton therapy as compared to photon'... change to: 'This maybe explained by the improved normal tissue dose distributions achieved with proton therapy due to the beam characteristics, as several dose comparison studies have observed the superiority of proton therapy as compared to photon therapy'
Done.
'However, defining the temporal distribution when the differences were significant is challenging. The change for xerostomia was found to be no longer significant after the acute phase [22], and the number of domains remained significantly worse for photon therapy after the acute phase was also reduced [24, 25]. In contrast, Cao et al. found the opposite, where the differences were only significant after 18 months'... change to: 'However, defining the temporal distribution when the differences were significant is challenging. Differences in xerostomia favouring PT during the first 3 months post treatment lost significance after the acute phase in one study [22], while a number of domains remained significantly worse for photon therapy beyond the acute phase in two other studies [24, 25]. In contrast, one further study. Cao et al. found differences between PT and photons were only significant after 18 months'
Done.
'First, the higher dose received from photon therapy...'.. change to 'First, the higher normal tissue doses received from photon therapy....'.
We have restructured this sentence taking into consideration comments from reviewers.
'First, a small number of the available study were conducted in a limited number of centres may limit the applicability of this review'..... change to 'First, there were only a small number of studies, which were conducted in a limited number of centres. This may limit the applicability of this review'
The sentence has been edited considering comments from reviewers.
Conclusion
'Proton therapy may improve patient-reported and quality-of-life outcomes for patients treated for oropharyngeal carcinoma, especially in acute and early late time points, which improve in later time points. This observation can be proven or challenged as more data available as proton therapy becomes more widely available.'.. change to: 'Proton therapy may improve patient-reported and quality-of-life outcomes for patients treated for oropharyngeal carcinoma, especially up to one year post-treatment. Symptoms may improve over time but may not return to baseline. The current evidence is, however, limited. As proton therapy becomes more widely available, it is hoped that further outcome data allow more definitive conclusions to be reached.'
The conclusion has been restructured considering all comments. It now reads:
Proton therapy may improve patient-reported and quality-of-life outcomes for pa-tients treated for oropharyngeal carcinoma, especially up to one year post-treatment. Symptoms may improve over time but may not return to baseline. The current evidence is, however, limited. As proton therapy becomes more widely available, it is hoped that fur-ther outcome data allow more definitive conclusions to be reached. The limitation high-lighted in the present study calls for collaboration between proton therapy centers to make multicenter randomized studies including carefully selected patients and more standard-ized treatment application.
Reviewer 3 Report
The authors performed a systematic review on the role of proton therapy in the treatment of oropharyngeal cancer, with respect to patient-reported outcomes. The topic is timely and relevant. The methodology is robust, and the manuscript is well-written. Few comments:
1) With respect to the PICOS criteria employed. P: patient population is patient treated with for oropharyngeal cancer. Any stage? Treated with which modality? Definitive RT + CHT? Surgery allowed?. Intervention is proton therapy. Any type? Setting is original research which is not a study design. Prospective/retrospective observational studies? Phase I-II-III trials? Other?
2) I suggest to expand PICOS to PICOTS and provide a time frame for the literature search.
3) How did authors reached agreement when disagreement is present with respect to paper selection and data extraction?
4) Any difference for surgical patients in the reported outcomes?
Author Response
Thank you for your insightful comments and suggestions.
The authors performed a systematic review on the role of proton therapy in the treatment of oropharyngeal cancer, with respect to patient-reported outcomes. The topic is timely and relevant. The methodology is robust, and the manuscript is well-written. Few comments:
1. With respect to the PICOS criteria employed. P: patient population is patient treated with for oropharyngeal cancer. Any stage? Treated with which modality? Definitive RT + CHT? Surgery allowed?. Intervention is proton therapy. Any type? Setting is original research which is not a study design. Prospective/retrospective observational studies? Phase I-II-III trials? Other?
P: patient population is patient treated with for oropharyngeal cancer.
Any stage? Yes. We do not specify which stage the patients are.
Treated with which modality? Definitive RT + CHT? The patients must be treated with proton therapy as a single modality or in combination with other treatments.
Surgery allowed? Yes.
Intervention is proton therapy. Any type? Yes.
Setting is original research which is not a study design. Prospective/retrospective observational studies? Phase I-II-III trials? Other? We do not specify the study design. Both prospective/retrospective in any phases were added.
2. I suggest to expand PICOS to PICOTS and provide a time frame for the literature search.
Thank you for your suggestion. We have changed the PICOS to PICOTS.
From our understanding, the T is related to the duration of treatment and the follow-up schedule that matter to patients. As we considered both long- and short-term outcomes, we accept studies with acute or late effects. We then divided them into acute, sub-acute, late (within one year), and late (after one year).
As for the literature search timeframe, we updated the timeframe from the first paper available in the databases till 3.2.2023, when we repeated the search.
- How did authors reached agreement when disagreement is present with respect to paper selection and data extraction?
Discrepancies in the selection results were deliberated in team meetings, where we went through the paper together to decide the suitability for inclusion.
- Any difference for surgical patients in the reported outcomes?
No study reported surgery as a significant clinical variable for the reported outcomes.
Reviewer 4 Report
This is a review about quality of life and patient-reported outcomes following proton therapy for oropharyngeal carcinoma. Seven reports were selected and included in the review.
The paper is well written. However, some issues remain.
The search strategy must be updated to 2023.
In table 3 the authors reported statistically significant outcomes but they did not report numeric results (e.g. percentages, mean scores). Please add them.
A table summarizing review results may help the readers.
Author Response
Thank you for the comments and suggestions
This is a review about quality of life and patient-reported outcomes following proton therapy for oropharyngeal carcinoma. Seven reports were selected and included in the review.
The paper is well written. However, some issues remain.
The search strategy must be updated to 2023.
Yes. The search has been repeated in 3.2.2023, and the PRISMA flowchart has been added, considering new articles found and assessed.
In table 3 the authors reported statistically significant outcomes but they did not report numeric results (e.g. percentages, mean scores). Please add them.
We have added the numeric results.
A table summarizing review results may help the readers.
We do not understand what you mean by this. We would be glad to add this table upon receiving clarification from you.
Round 2
Reviewer 1 Report
Dear authors,
The article has been carefully revised.
I have no further comments for improvement.
Best regards
Author Response
Thank for for your feedback.
Reviewer 4 Report
Thanks for improving the manuscript.
Author Response
Thank you for your feedback.